# Improved Leakage Detection and Recognition Algorithm for Residual Neural Networks Based on Transfer Learning

**Liangliang Li [1], Yu Chen [1,2,3,\*], Zhengxiang Ma [1,2], Xinling Wen [1,2,3], Jiabao Pang [1] and Weitao Yuan [1]**

1    School of Intelligent Engineering, Zhengzhou University of Aeronautics, Zhengzhou 450046, China;
     lllliang@zua.edu.cn (L.L.); mzx@zua.edu.cn (Z.M.); wenxinling@zua.edu.cn (X.W.); pjone@zua.edu.cn (J.P.);
     yweitao0520@zua.edu.cn (W.Y.)
2    Collaborative Innovation Center of Aeronautics and Astronautics Electronic Information Technology,
     Zhengzhou University of Aeronautics, Zhengzhou 450046, China
3    Henan Key Laboratory of General Aviation Technology, Zhengzhou University of Aeronautics,
     Zhengzhou 450046, China
*    Correspondence: chenyu@zua.edu.cn; Tel.: +86-037161912109

**Abstract:** Due to the lack of other component information in traditional magnetic leakage signal defects and the low accuracy of prediction methods, this paper proposes an improved residual network for magnetic leakage detection defect recognition method that predicts defect size and different detection speeds. A new defect diagnosis method based on ResNet18 on the Convolutional Neural Network (CNN) is proposed in this study. This method transfers the pre-trained ResNet18 network and replaces the activation function in the transferred network structure. It extracts features from transformed two-dimensional images obtained by converting the original experimental signals and signals with added noise, removing the influence of manual features. The results demonstrated that the improved ResNet18 network model, after transfer learning, achieved 100% prediction accuracy for all 10,000 grayscale images generated with defect lengths of 50 mm; width of 2 mm; and depths of 2 mm, 4 mm, 6 mm, and 8 mm. Moreover, the prediction accuracies for the quasi-static, slow, compensated fast, and fast scanning speeds were 99.20%, 98.50%, 93.30%, and 94.00%, respectively, for defect depths of 2 mm, 4 mm, 6 mm, and 8 mm. These accuracies surpass those of other models, demonstrating the significant improvement in prediction accuracy achieved by this method.

**Keywords:** magnetic leakage signal; defect detection; residual network; transfer learning

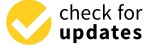



## 1. Introduction

Steel, which is primarily composed of ferromagnetic materials, is the industry's "staple food". It is widely used in various engineering fields such as pipelines, steel plates, construction, transportation, and aerospace due to its high strength, hardness, corrosion resistance, plasticity, and toughness. As a result, ferromagnetic materials are vital for national industrial and military development. Pipelines play an important role in energy systems [1]. They are used to transport natural resources such as oil and gas. Pipelines and other ferromagnetic materials are subjected to a variety of damage challenges in various environments, including electrochemical reactions in the environment, welding defects, and external force damage. Defects such as metal loss, pitting, and cracking can jeopardize pipeline integrity [2], resulting in pipeline accidents. Not only these accidents cause significant economic losses, but also they pose the risk of landslides, fires, explosions, or pollution [3]. Regular internal inspections are usually performed to assess defects and other issues in order to ensure the safe use of in-service ferromagnetic materials. Visual inspection of the surface of ferromagnetic materials is the simplest and most widely used inspection method, but it requires a large amount of manpower and resources and is very expensive. It must also be restored after inspection [4]. Non-destructive testing is an important quality-control tool in heavy industries such as oil and gas [5]. Magnetic flux

leakage (MFL) [6,7] is a common non-destructive testing method that involves magnetizing ferromagnetic materials in the inspected pipeline with magnets and collecting leakage magnetic field signals near the defects to estimate the defect size [8], allowing pipeline safety assessment.

Currently, domestic and international researchers primarily predict the size of pipeline defects by extracting characteristic values from the axial leakage component and radial leakage component in leakage detection data from tested pipeline materials [9]. Manual defect size recognition takes a long time and is very labor-intensive. Currently, the accuracy of pipeline defect recognition is largely dependent on the decisions of experienced engineers. However, this method is primarily based on a single-dimensional leakage magnetic signal feature, with no other dimensional feature information, and signal feature information extraction is limited, resulting in unsatisfactory defect prediction and recognition effects.

In recent years, convolutional neural networks have experienced rapid development, breaking the limitations of manual features. Manual features refer to features designed and selected by humans for object recognition and classification. However, manual feature extraction has some limitations, including the need for a large amount of time and expertise, as well as limited expressive power. The CNN model is utilized for feature extraction, enabling the extraction of higher-dimensional features. Here, high-dimensional features refer to feature vectors with a large number of elements or indicators. Compared with low-dimensional features, high-dimensional features can provide more information and richer representation capabilities and have been widely used in the field of computer vision. Some important algorithms are based on this, such as Visual Geometry Group Network (VGGNet) [10], Long Short-Term Memory (LSTM) [11], Faster Region Convolutional Neural Network (Faster R-CNN) [12], and You Only Look Once (YOLO) [13]. These models can automatically extract image features and achieve good results on publicly available international datasets. The basis of magnetic leakage detection is the principle of magnetic leakage. Magnetic leakage refers to the magnetic field leaked into the external space in a specific magnetic circuit. In the magnetic leakage detection of ferromagnetic material defects, the principle is shown in Figure 1. Permanent magnets are used to excite ferromagnetic materials, forming a closed magnetic circuit between the laminations, air gaps, and ferromagnetic material. After saturation excitation of ferromagnetic materials, the magnetic induction intensity inside the non-defective ferromagnetic material remains stable; that is, the magnetic lines are uniformly distributed inside the ferromagnetic material. However, the permeability sharply decreases at the defect, and at this time, the magnetic lines will overflow from the surface of the ferromagnetic material defect, forming a magnetic leakage field in the air on the surface of the ferromagnetic material. The intensity and distribution of the magnetic leakage field are directly related to the excitation intensity, defect type, and defect size. When the excitation intensity is determined, the damage can be quantitatively identified based on the distribution of the magnetic leakage field. By using magnetic sensitive elements to obtain the above magnetic leakage field signals, the width, depth, and other features of the corresponding defects can be extracted. This paper proposes a magnetic leakage detection and identification method based on an improved deep convolutional neural network. It converts the one-dimensional data features of axial and radial magnetic leakage into two-dimensional grayscale images, which are used as inputs to the neural network. Based on transfer learning technology [14], the pre-trained improved deep convolutional neural network model is used to fuse the magnetic leakage defect data features for feature extraction [15], achieving defect regression classification.

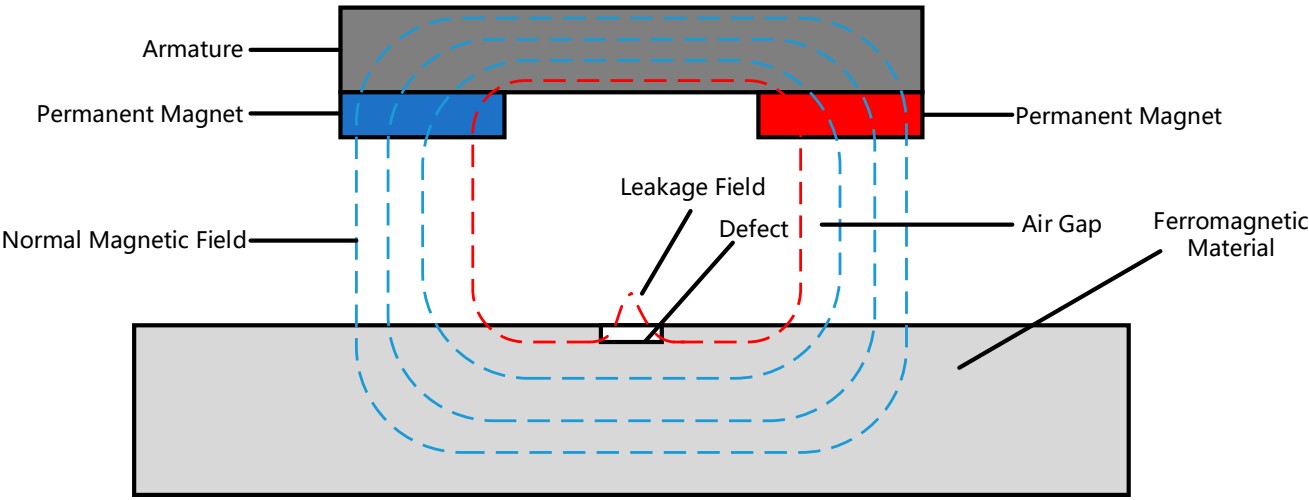

**Figure 1.** Detection principle of magnetic leakage method.

## 2. Basic Principles

### 2.1. Two-Dimensional Magnetic Leakage Signal

In order to investigate the influence of defect depth on the magnetic flux leakage signal, the magnetic dipole model of the rectangular defect model was adopted for analysis. Assuming the mass of the rectangular defect model is zero, the radial component of the magnetic flux leakage field (the component of the magnetic field in the normal direction to the surface of the object) can be expressed as follows:

$$H_x = \frac{\rho_s}{2\pi\mu_0}\left(arctan\frac{h+y}{n-x} + arctan\frac{h+y}{n+x} - arctan\frac{y}{n-x} - arctan\frac{y}{n+x}\right) \tag{1}$$

The axial component of the leakage magnetic field (the component of the magnetic field in the tangential direction on the surface of the object) is

$$H_y = \frac{\rho_s}{2\pi\mu_0}\log\left[\frac{(n+x)^2 + (h+y)^2}{(n+x)^2 + y^2}\frac{(n-x)^2 + y^2}{(n-x)^2 + (h+y)^2}\right] \tag{2}$$

In the equation, if the charge density $\rho_s$ remains constant, $\rho_s/2\pi\mu_0$ can be regarded as a constant, so only the defect depth, defect width $n$, and distance $y$ affect the distribution of the leakage magnetic field at the defect location. According to the principle of magnetic leakage detection, the patterns of magnetic leakage signals with the same length and width but different depths, as well as their radial and axial components, are shown in Figure 2. When the length and width of defects are equal, a larger depth leads to a higher peak value of the detected leakage magnetic signal, as well as higher peak values of the decomposed radial and axial components. The valley spacing is a defect characteristic value. Taking the defect width $n$ as 5 mm and maintaining a separation distance of 5 mm, the defect depth $h$ is simulated starting from 2 mm and increasing by 2 mm increments up to 8 mm, as shown in Figures 3 and 4. From the figures, it can be seen that, under the same conditions, changing the defect depth will change the peak-to-peak values of the axial and radial data [16].

### 2.2. Convolutional Neural Network

The convolutional neural network is a well-known deep-learning method. In order to improve the classification performance, the size of the images in this study varies according to the amount of signal data, and larger image sizes can improve the classification results. This study has a large amount of signal data, and when the generated grayscale image size is set to 64 × 64, a total of 10,000 grayscale images can be generated, which can meet the requirements of convolutional neural network training. Therefore, setting the input image

size to 64 × 64 is most appropriate. As a comparative model, a simple CNN model with three convolutional layers and two fully connected layers is built in this paper, as shown in Figure 5.

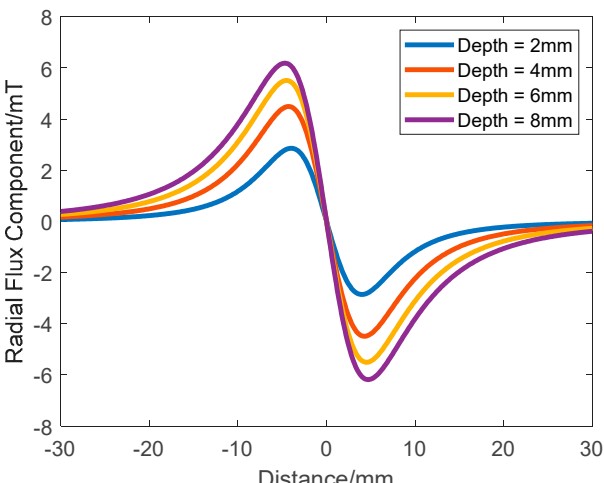

**Figure 2.** The characteristics of defect leakage magnetic signals in the radial and axial components.

**Figure 3.** Radial leakage curve.

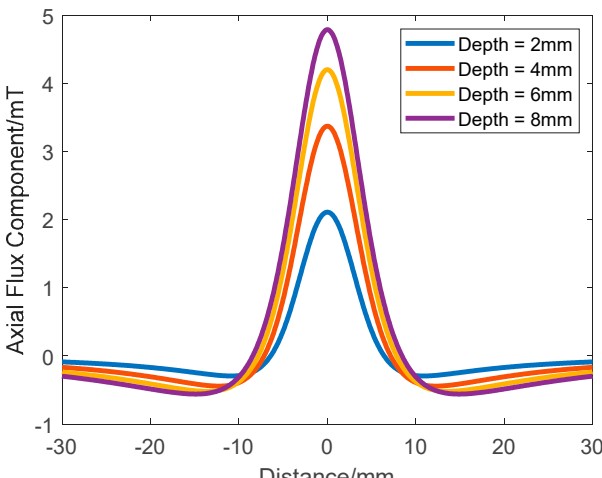

**Figure 4.** Axial leakage curve. Note: $\rho_s/2\pi\mu_0$ is the constant, defect width $n$ = 5 mm, lift-off distance $y$ = 5 mm, and defect depth $h$ is taken as 2 mm, 4 mm, 6 mm, and 8 mm, respectively. Under the same conditions, as the defect depth increases, the difference between the peak and valley values of the radial leakage magnetic signal data will increase, and the peak value of the axial leakage magnetic signal data will increase.

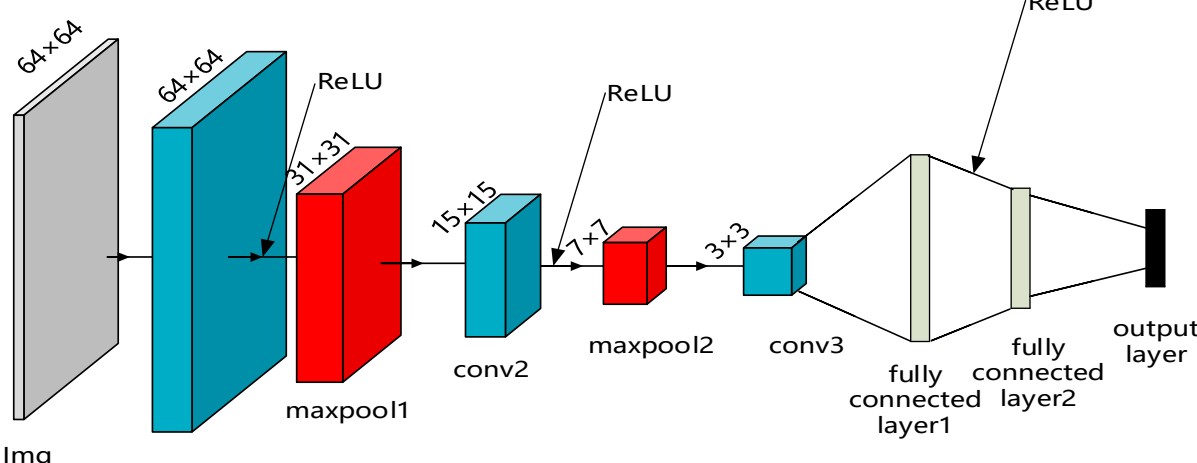

**Figure 5.** Simple CNN architecture.

The input of this model is image data of size $64 \times 64$ with 3 channels, and the convolutional layers with 16 output channels perform convolution operations using $3 \times 3$ convolution kernels. Then, a nonlinear mapping is performed through the Rectified Linear Unit (ReLU) activation function, and spatial downsampling is also performed through a $2 \times 2$ max pooling layer to reduce the size of the feature map. This process is repeated three times, with each iteration increasing the number of output channels to 16, 32, and 64, respectively, in order to extract features of different scales and complexities. The resulting feature maps are then transformed into one-dimensional vectors through the flattening operation. Finally, classification prediction is performed through two fully connected layers. The description is as follows:

(1)    Feature extraction section:

The first convolutional layer: The first convolutional layer of the feature extractor part uses a convolutional kernel from 3 channels to 16 channels and obtains 16 feature maps by convolving the input image. The size of the convolutional kernel is $3 \times 3$ with a stride of 2. This reduces the size of the input image by half compared with the original size.

ReLU activation function layer: Apply the output of the convolutional layer to the ReLU activation function to introduce nonlinearity and increase the expressive power of the network.

Max pooling layer: Perform max pooling operation on the feature map using a $2 \times 2$ window and 2 strides, downsampling the feature map and reducing its size and parameter count.

The second convolutional layer: The input channel number is 16, the output channel number is 32, the convolution kernel size is $3 \times 3$, and the stride is 2. This will further reduce the size of the feature map by half.

ReLU activation function layer: By subjecting the output of the convolutional layer to the ReLU activation function, nonlinearity is infused into the network, enhancing its expressive prowess.

Max pooling layer: By utilizing a $2 \times 2$ window with 2 strides, execute the max pooling operation on the feature map. This downscales the feature map, simultaneously decreasing its size and parameter count.

The third convolutional layer has an input channel of 32, an output channel of 64, a kernel size of $3 \times 3$, and a stride of 2. This convolutional layer further reduces the size of the input feature map by half.

ReLU activation function layer: Employing the ReLU activation function on the output of the convolutional layer introduces nonlinearity, thereby augmenting the network's expressive capability.

Max pooling layer: Perform max pooling operation on the feature map using a $2 \times 2$ window and 2 strides, downsampling the feature map and reducing its size and parameter count. At this point, we obtained feature maps with 64 channels, and their spatial dimensions have been greatly reduced. These feature maps are flattened into a vector and input into the classifier for classification.

The feature extraction part uses two similar convolutional layers and pooling layers in sequence, and each convolutional layer uses convolution kernels from the output channel number of the previous layer to the channel number of the current layer. The purpose of doing this is to gradually extract more abstract and advanced features through multiple layers of convolutional operations.

Through these convolutional layers and pooling layers, the input image undergoes multiple downsampling while extracting higher-level features.

(2)  Classifier section:

It consists of two linear layers. First, the feature maps outputted by the feature extraction layer are flattened into a one-dimensional vector through a linear layer with 576 input features and 256 output features. Then, through a linear layer with 256 input features and num_classes output features (num_classes is the number of classification categories), the feature vector is mapped to the final classification result. In this model, the ReLU activation function is used for nonlinear mapping between two linear layers. The description is as follows:

The first fully connected layer: The input size is 576 (the size of the flattened feature map from the previous step), and the output size is 256. This fully connected layer converts the feature map into a higher-dimensional feature representation.

The second fully connected layer: The input size is 256, and the output size is num_classes (the number of categories). This fully connected layer maps the final feature representation to the dimension of the number of classes for classification prediction.

During the forward propagation process, the input data first undergo feature extraction, where the convolutional layers and pooling layers extract the features of the images. Then, the feature maps are flattened into a one-dimensional vector through the flattening operation. Finally, the unfolded feature maps are mapped to class scores through the fully connected layer of the classifier section.

### 2.3. ResNet18 Network Model Improved Based on Transfer Learning

Due to the need for a large amount of data for training deep-learning models, it is difficult to train a practical network model with limited actual experimental data. Transfer learning can reduce the dependence of neural network models on the amount of data. Therefore, the pre-trained ResNet18 network model [17] is used, and the network structure and parameters of the pre-trained model are transferred to the improved neural network model through transfer learning, thereby achieving the prediction of magnetic leakage defect size in ferromagnetic materials.

In transfer learning, further optimization of the ResNet18 network model includes improving the ReLU activation function to Leaky Rectified Linear Unit (LeakyReLU) function in the original framework, importing pre-trained weights and parameters into the convolutional layers, forming a new convolutional neural network, and finally training the new model with experimental data. The improved model is shown in Figure 6.

(1) LeakyReLU function

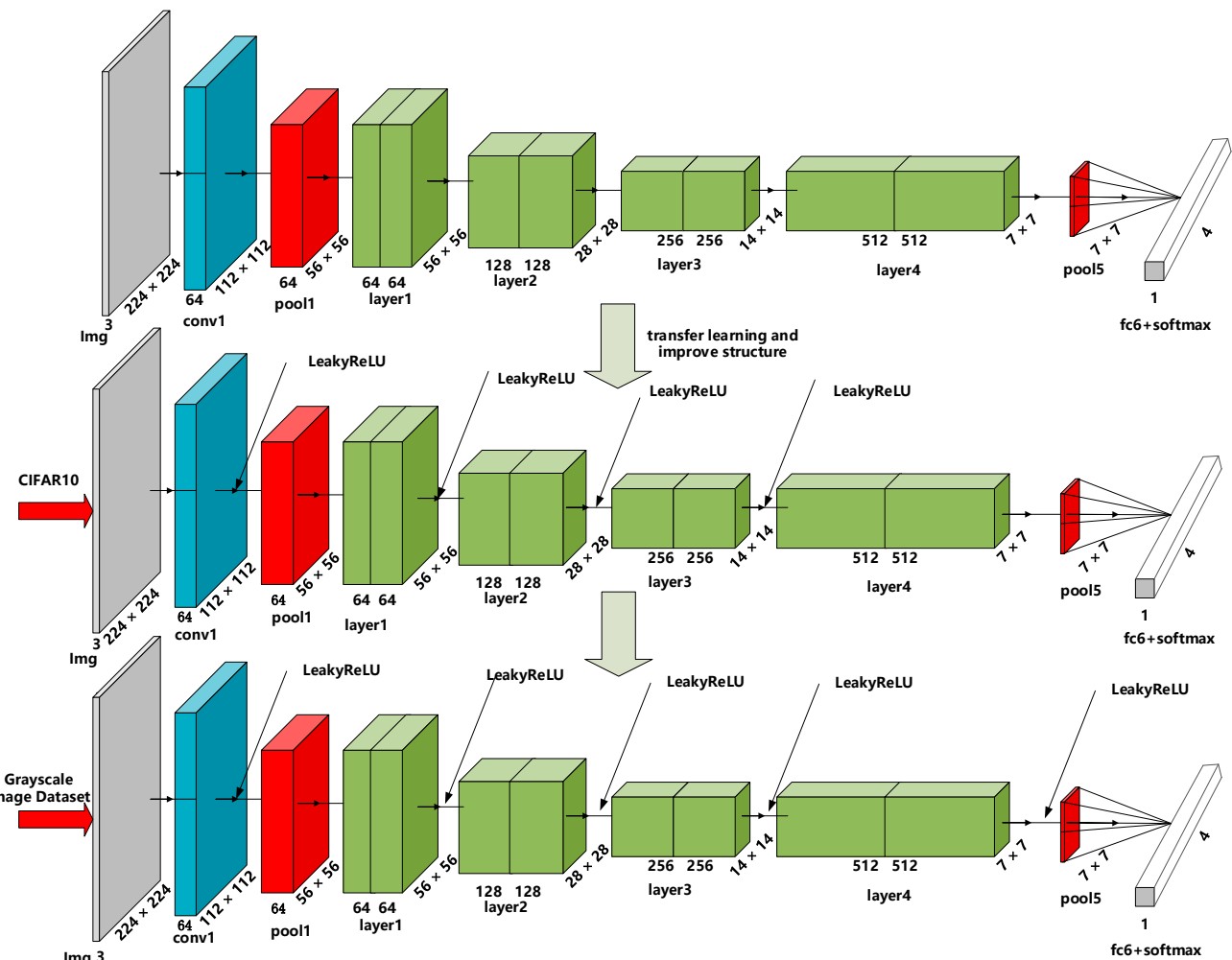

**Figure 6.** Improved ResNet18 architecture with transfer learning.

LeakyReLU is a variant of the rectified linear unit used in the design of activation functions. Avoiding "neuron death" and the vanishing gradient problem ensures information flow. Its function is

$$\mathrm{R}:\left(y_j^l\right)=\max\left(0, Y_j^l\right) \tag{3}$$

$$\mathrm{LR}:\left(y_j^l\right)=\max\left(0, Y_j^l\right)+\mathrm{a}*\min(0, Y_j^l) \tag{4}$$

where a is a small parameter, usually taken as a small positive number, which retains a portion of the negative axis data so that the negative axis data are not completely lost.

(2)　Objective function

The cross-entropy loss is used as the accuracy metric for regression prediction models, as shown in Equation (5).

$$\text{Loss} = -(\frac{1}{n}) * \sum (y * \log(y_{\_pred}) + (1 - y) * \log(1 - y_{\_pred}) \tag{5}$$

where n is the sample size, $y_{\_pred}$ is the predicted defect size value of the model, and y is the true defect size value.

(3)　Training algorithm

Stochastic Gradient Descent with Momentum (SGDM) is used to train the network. Each time, a random sample is chosen from the training set for learning, and the exponential weighted average of the gradient is calculated to update the Kij and bj of the sampled data.

$$K_{ij} = K_{ij} - \eta v_K(t)\frac{\partial E}{\partial K_{ij}} \tag{6}$$

$$b_j = b_j - \eta v_b(t)\frac{\partial E}{\partial K_{ij}} \tag{7}$$

$$v_K(t) = -\eta\frac{\partial E}{\partial K_{ij}} + v_K(t-1) * momentum \tag{8}$$

$$v_b(t) = -\eta\frac{\partial E}{\partial b_{ij}} + v_b(t-1) * momentum \tag{9}$$

In the equation, $\eta$ is the learning rate of the CNN; $v_K(t)$ and $v_b(t)$ are the momentum of the two parameters during the t-th learning; and momentum $\in [0, 1]$.

## 3. A Transfer Learning Improved ResNet18 Network Fault Diagnosis Method Based on Two-Dimensional Signals

The specific process includes data preprocessing, transfer learning, and optimization of the ResNet18 network model. By obtaining magnetic leakage signal data, preprocessing the data, that is, augmenting the original data and overlaying the data with reasonable Gaussian noise, the dataset is expanded. The radial signal data and axial signal data of the magnetic leakage data are fused together and transformed into a two-dimensional image. Then, the ResNet18 model is transferred and improved by replacing all activation functions with LeakyReLU functions. The overall framework of the algorithm is shown in Figure 7.

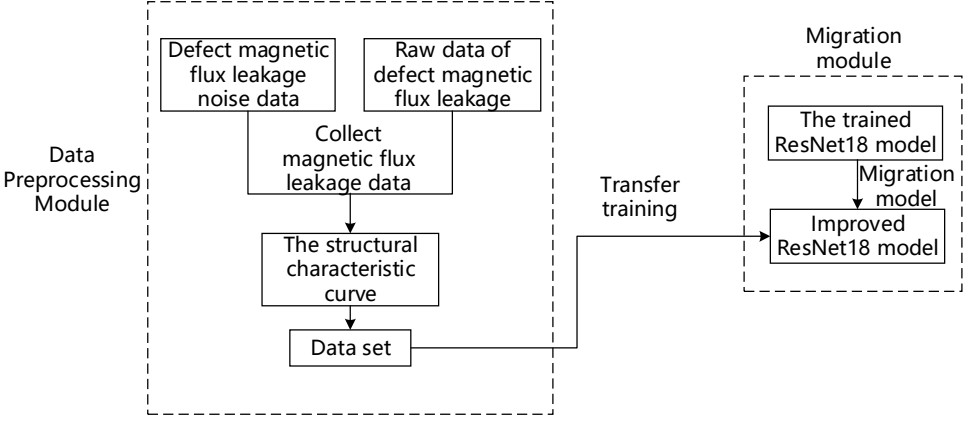

**Figure 7.** Algorithm overall flowchart.

### 3.1. Data Preprocessing

Adding various noises to the leakage magnetic data, normalizing the leakage magnetic defect data, and transforming the leakage magnetic defect data into the range of 0–1 [18], in order to obtain an M × M size image, it is necessary to fuse the radial data and the axial leakage magnetic feature data together, as shown in Figure 8.

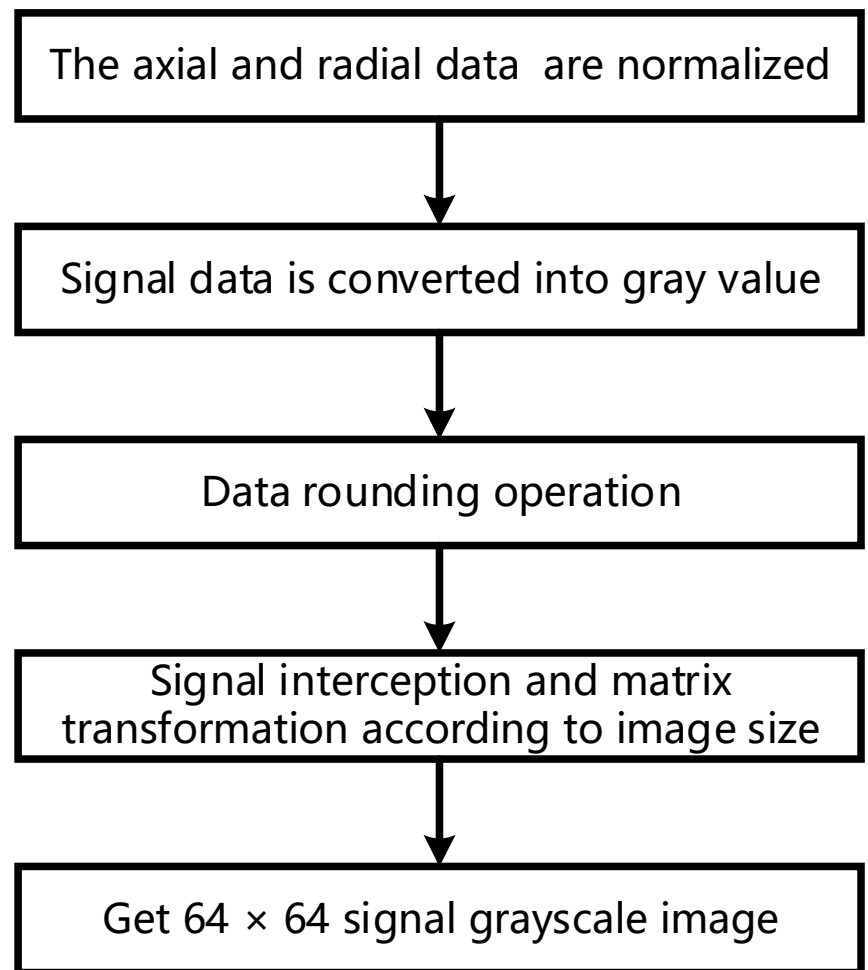

**Figure 8.** Process flowchart of converting magnetic signals to grayscale images.

Obtain a segment signal of length $M^2$, and let L(i) represent the segmented signal, where i = 1, 2, 3... $M^2$, P (o, p) represents the pixel intensity of the image, where o = 1, 2, 3... M, p = 1, 2, 3... M, as shown in Equation (10).

$$P(o, p) = F\{\frac{L((o-1) * M + p) - Min(L)}{Max(L) - Min(L)} * 255\} \tag{10}$$

The function F{} is a rounding function, and the entire pixel value has been normalized from 0 to 255, which is only the pixel intensity of the grayscale image.

### 3.2. Signal to Grayscale Image Conversion

Convert one-dimensional leakage magnetic data into two-dimensional grayscale images. According to the principle of leakage magnetic technology, the two sets of data are cross-fused for output. Then, set the size to 64 × 64 pixels, as shown in Figure 9.

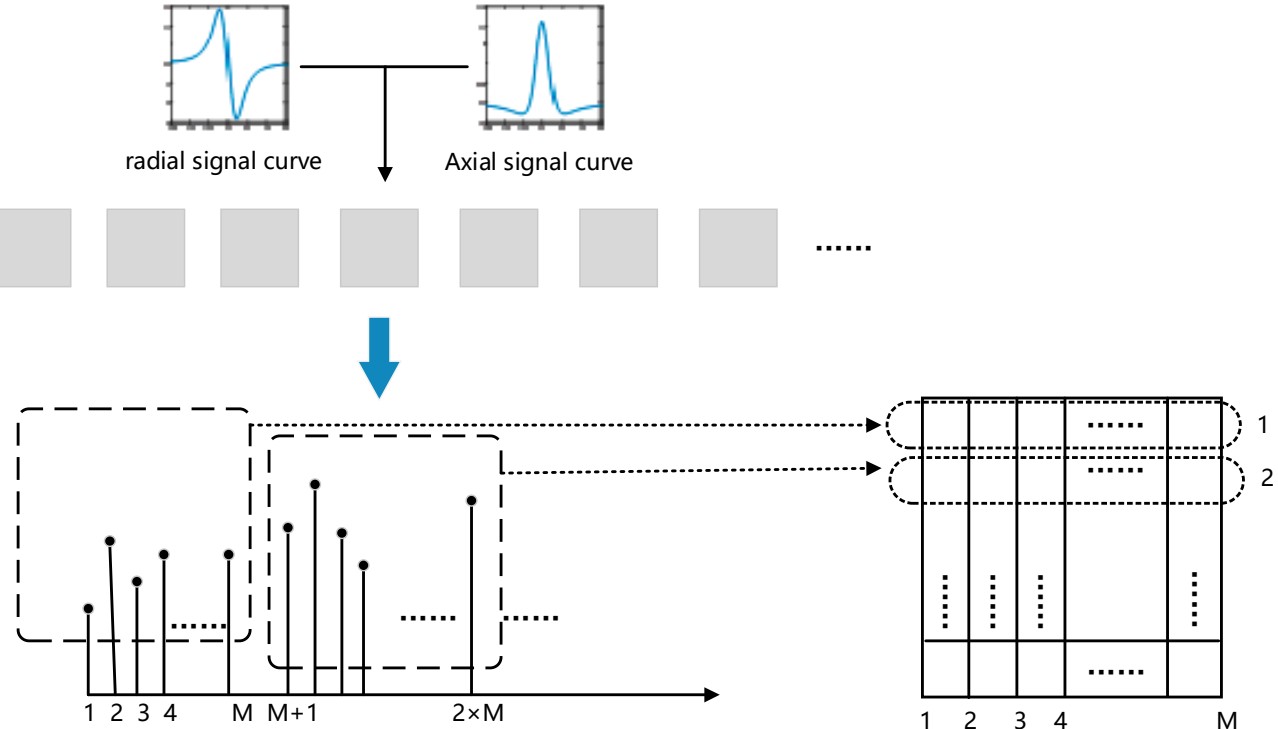

**Figure 9.** Signal–image conversion diagram.

A dataset of pipeline leakage magnetic curve images based on convolutional neural networks is established, with each type of defect featuring image pixels sized at $64 \times 64$. Figure 10 represents randomly selected grayscale images associated with different defect sizes, specifically four grayscale images with defect lengths of 50 mm; width of 2 mm; and depths of 2 mm, 4 mm, 6 mm, and 8 mm. From the figure, it can be observed that as the defect depth increases, the proportion of the lighter-colored areas in the feature part decreases, resulting in darker shades in the feature images.

*3.3. Optimization of ResNet18 in the Migration Process*

This article generates a specific model by calling the resnet18() function, and the resnet18 function constructs the network using the ResNet class. The forward() function in the ResNet class specifies the flow of network data.

(1)   After the data enter the network, they first go through the input part (conv1, bn1, relu, maxpool);

(2)   Then the data enter the intermediate convolutional part (layer1, layer2, layer3, layer4);

(3)   Finally, the data are passed through an average pooling and fully connected layer (avgpool, fc) to obtain the result.

1.   Network input optimization

The input part of the ResNet18 network consists of a large convolutional kernel with size = $7 \times 7$ and stride = 2, as well as maximum pooling with size = $3 \times 3$ and stride = 2. Through this step, an input image of size $224 \times 224$ will be transformed into a feature map of size $56 \times 56$, greatly reducing the required storage size. The modification in the input layer of the network in this paper is to replace the Relu activation function with the LeakyReLU activation function, and the structural modification is shown in Figure 11.

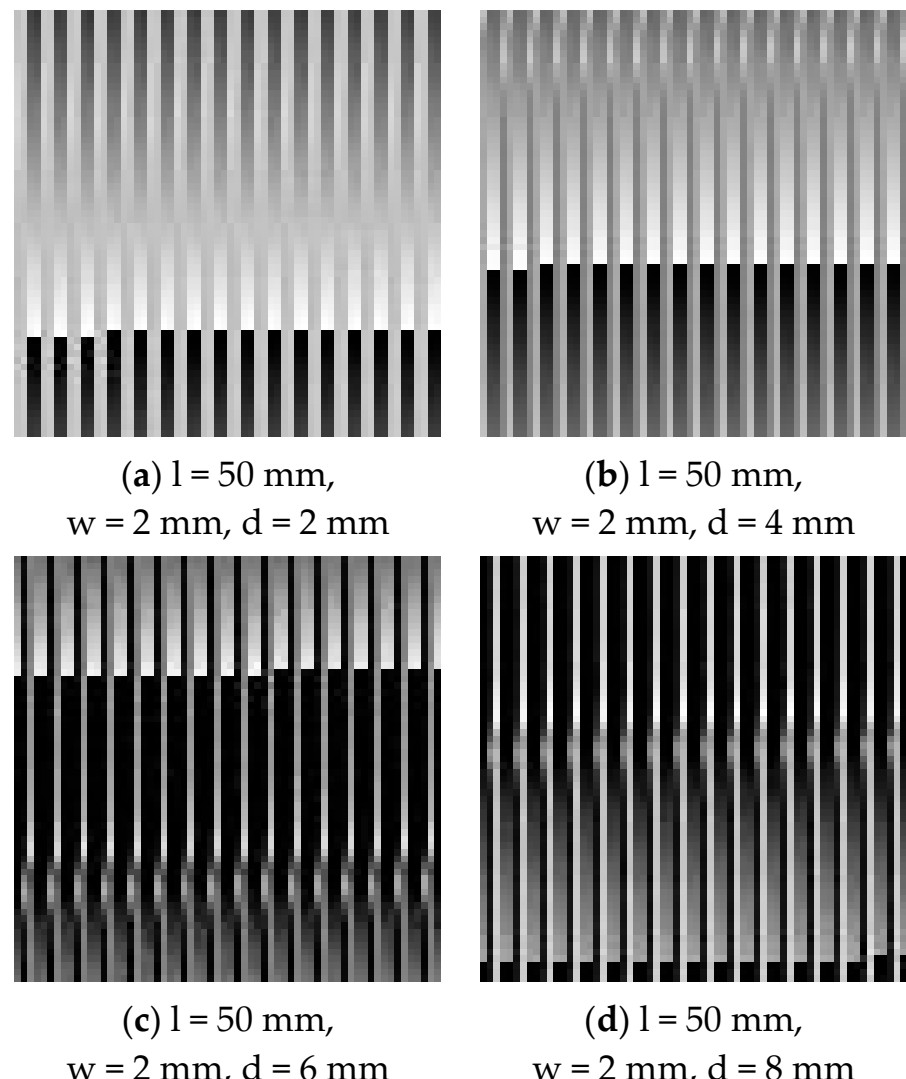

(**a**) l = 50 mm,
w = 2 mm, d = 2 mm

(**b**) l = 50 mm,
w = 2 mm, d = 4 mm

(**c**) l = 50 mm,
w = 2 mm, d = 6 mm

(**d**) l = 50 mm,
w = 2 mm, d = 8 mm

**Figure 10.** Feature maps corresponding to different defect sizes. "l" represents the length of the defect, "w" represents the width of the defect, and "d" represents the depth of the defect.

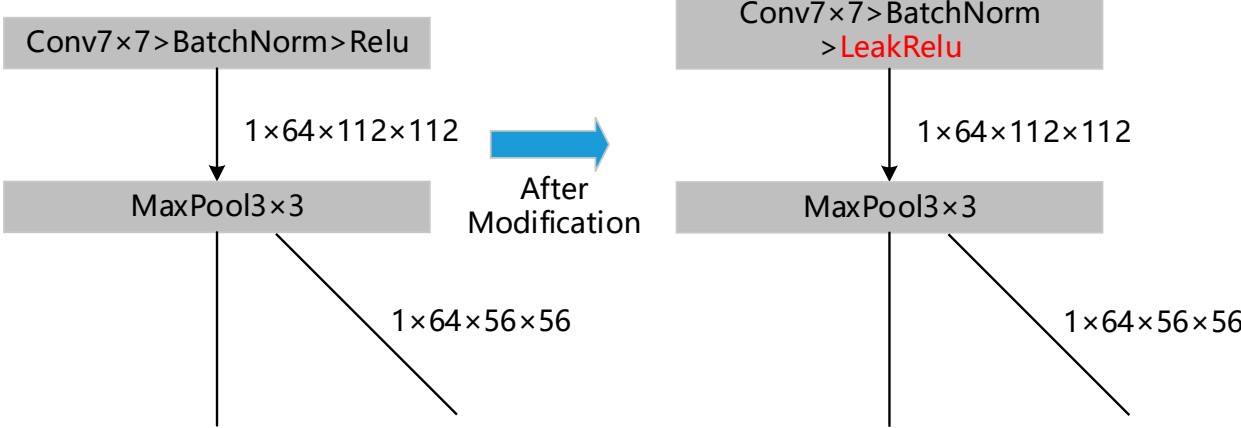

**Figure 11.** Input layer feature changes.

2. Optimization of intermediate convolutional layers in the network

In ResNet18, each residual block consists of 2 Basic Blocks. Basic Block is the basic unit that constitutes a residual block. ResNet18 forms different layers of residual blocks by stacking multiple Basic Blocks. The modification to the ResNet18 model is to replace the ReLU activation function in the Basic Block of the 4 layers with the LeakyReLU activation function. The specific structural change is shown in Figure 12.

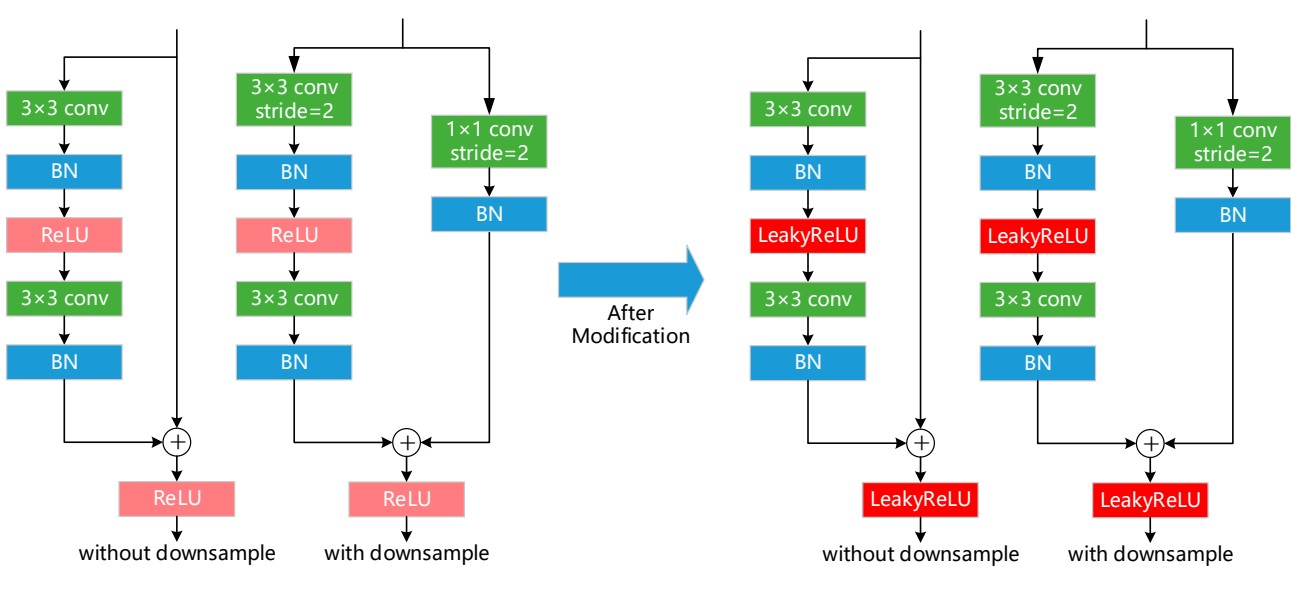

**Figure 12.** Basic Block changes.

## 4. Algorithm Verification

### 4.1. Data Collection

This article uses the MFL tool to scan the test data of the test plate made of S355 steel grade [19]. That is, select the detection data of 2 mm, 4 mm, 6 mm, and 8 mm depth defects and select the quasi-static, slow, compensated fast, and fast four detection speeds for the same type of defects. The output of sensor Bx is directly proportional to the radial (horizontal) component of MFL, while the output of sensor Bz1 is directly proportional to the axial (vertical) component of MFL. Therefore, the voltage values output by sensor Bx and Bz1 can be used instead of radial magnetic leakage signal data and axial leakage magnetic data. The measurement platform consists of three magnetizers, one digital encoder, and one measurement module mentioned in the above article. It uses a linear Hall effect sensor A1324 to measure the magnetic flux leakage. After obtaining the original experimental data, this paper adds Gaussian noise to the original data and finally obtains a dataset with 10,000 defective samples. The dataset is randomly divided into a training set, a validation set, and a test set in a ratio of 8:1:1. The neural network model's program is written in Python 3.7.0 and uses the open-source framework Pytorch. The model training is performed using GPU with a batch size of 64 and 2500 iterations.

### 4.2. Model Evaluation Index

The experimental results in this paper use accuracy, precision, F1, and $R^2$ values as the evaluation metrics for the models. Accuracy is the most important evaluation criterion for model classification results in intrusion detection technology, and the other three metrics have also been widely used by researchers to evaluate intrusion detection models. Among them, the larger the accuracy, precision, F1, and $R^2$ values, the better the performance of the model.

In the confusion matrix of binary classification, the four elements represent the following meanings (Table 1):

**Table 1.** Confusion Matrix for Classification Results.

| Actual Category | Predicted Category | |
|---|---|---|
| Anomaly or normal | Anomaly | Normal |
| Anomaly | TP | FN |
| Normal | FP | TN |

TP (true positive): the number of positive samples correctly classified by the intrusion detection model in abnormal behavior;

FN (false negative): the number of negative samples misclassified by the intrusion detection model in abnormal behavior;

FP (false positive): The number of positive samples that are incorrectly classified as anomalies by the intrusion detection model in abnormal behavior;

TN (true negative): the number of negative samples correctly classified by the intrusion detection model in abnormal behavior.

In this article, the main reasons for the occurrence of FN are the addition of Gaussian noise to the data and the too fast detection rate, which causes data variation and increases outliers in the data, leading to the model incorrectly classifying positive samples as negative samples, resulting in false negatives.

(1) Accuracy:

$$\text{Accuracy} = \frac{\text{TP} + \text{TN}}{\text{TP} + \text{FP} + \text{TN} + \text{FN}} \tag{11}$$

Accuracy refers to the proportion of behaviors correctly classified by the intrusion detection model among all behaviors, as shown in Formula (11).

(2) Precision:

$$\text{Precision} = \frac{\text{TP}}{\text{TP} + \text{FP}} \tag{12}$$

Precision refers to the proportion of true-positive samples in all intrusion behaviors in the model classification, which means the proportion of correctly detected abnormal attacks in the predicted abnormal attack types. It is calculated using Formula (12).

(3) F1-score value:

$$\text{F1} = \frac{2 \times \text{Precision} \times \text{Recall}}{\text{Precision} + \text{Recall}} = \frac{2 \times \text{TP}}{2 \times \text{TP} + \text{FN} + \text{FP}} \tag{13}$$

F1 (F1-score) is a judgment criterion that combines accuracy and recall. It is the harmonic mean of these two values and comprehensively considers both precision and recall. The larger the value, the better the model's performance, as shown in Formula (13).

(4) $R^2$ (coefficient of determination):

$R^2$, also known as the coefficient of determination, is a statistical indicator used to evaluate the fitting degree of a regression model. It represents the explanatory power of the regression model on the observed data. The value range of $R^2$ is between 0 and 1. The closer it is to 1, the better the model fits the data, while the closer it is to 0, the worse the model fits the data.

### 4.3. Verification Results of Different Defect Depths

Using the improved ResNet18 after migration for recognition, selecting a suitable learning rate $L_R = 0.01$ for the network, selecting the cross-entropy loss function CrossEn-

tropyLoss(), and selecting the momentum of stochastic gradient descent $M_T = 0.9$, the training results are shown in Figure 13.

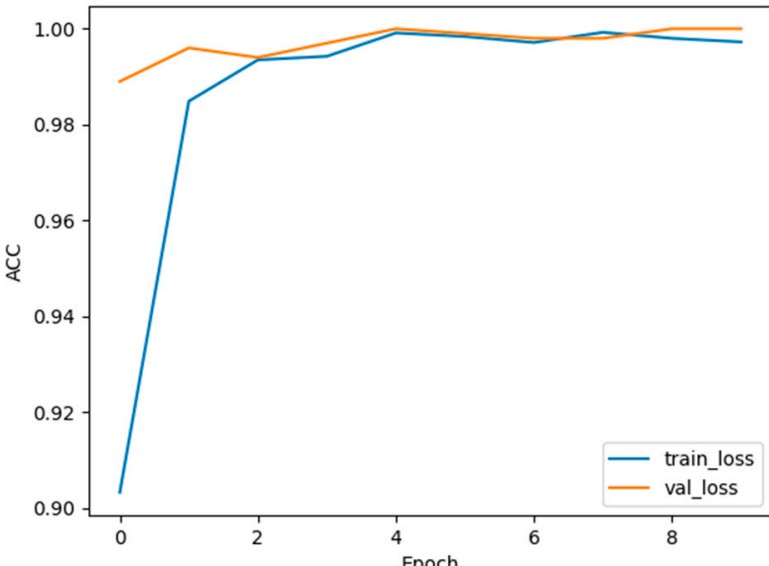

**Figure 13.** Identification accuracy of ResNet 18 after migration improvement.

In order to verify the advantages of the improved ResNet18 model after migration, the non-migrated ResNet18 model, the migrated ResNet18 model, and the CNN model were established. Select neural network hyperparameters, i.e., $L_R = 0.01$, $M_T = 0.9$. The confusion matrices of the four models for the four defects are shown in Figure 14. The evaluation metrics of the four network models for the four defects are shown in Table 2. The results show that the improved ResNet18 defect recognition performs the best.

### 4.4. Verification Results of Different Scanning Speeds

During magnetic leakage detection, the scanning detection speed can also affect the accuracy and reliability of the detection. Higher scanning detection speed can improve scanning detection efficiency, especially for large pipelines or situations requiring long detection time. This helps to reduce detection time and workload and improve productivity. Lower detection speed can increase the sensitivity to subtle defects. When the detection speed is slow, the detector has more time to perceive smaller magnetic changes. This is very beneficial for detecting small cracks or corrosion defects. Lower detection speed helps ensure data quality. When the scanning detection speed is slow, the scanning detection equipment can collect more data points, thus providing more accurate and reliable results.

Four different scanning speeds, quasi-static, slow, compensated fast, and fast, were selected from the detection data under the reference literature. Gaussian noise was added to the data to augment it for training purposes. Then, for training, the improved migrated ResNet18 model, migrated ResNet18 model, non-migrated ResNet18 model, and CNN model are used, with a comparison of the downward trend of their cost functions shown in Figure 15. The cost function is the average of all sample errors, and it is used to calculate the difference between the model's predicted and actual results. It is also known as the loss function or the objective function. The cost function in this paper is cross-entropy. Cross-entropy can quantify the difference between the model output and the actual label and can provide useful gradient information during the model optimization process, allowing the model to update its parameters quickly and effectively. The value of cross-entropy will be very large if the model's classification result is very different from the actual label, whereas the smaller the difference between the prediction result and the label, the smaller the value of cross-entropy; that is, the smaller the cost function value, the better the fitting of the trained model.

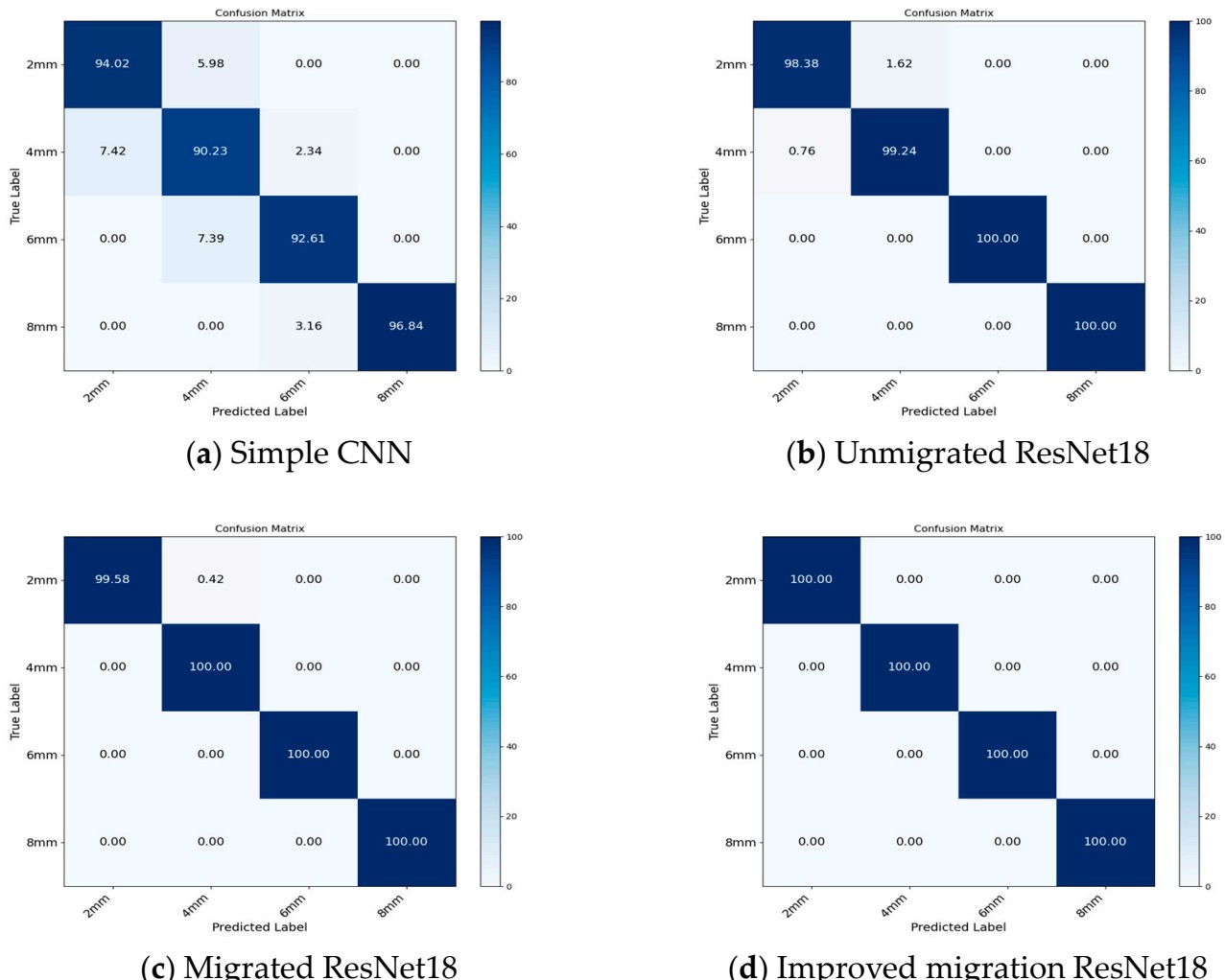

**Figure 14.** Comparison of confusion matrices for four types of defects.

**Table 2.** Comparison of evaluation metrics for defect recognition in different network models.

| Model | Accuracy | Precision | F1 | $R^2$ |
|-------|----------|-----------|-----|-------|
| Improved migration ResNet18 | 100.0% | 100.0% | 100.0% | 100.0% |
| Migrated ResNet18 | 99.9% | 99.9% | 99.9% | 99.9% |
| Unmigrated ResNet18 | 99.4% | 99.4% | 99.4% | 99.5% |
| Simple CNN | 93.4% | 93.5% | 93.4% | 94.6% |

By minimizing the cost function, during the training process, the parameters of the model will gradually adjust to the state where the cost function value is minimized, so that the training process converges to a relatively good model state. Therefore, the selection and optimization of the cost function are very important for training an accurate and high-performance convolutional neural network model. From Figure 15 and Table 3, it can be seen that when the training sample image format is 64 × 64, the improved ResNet18 model with transfer learning has the same cost function value as the transfer ResNet18 model in detecting 2 mm defects. The cost function value is the smallest when detecting 4 mm, 6 mm, and 8 mm defects. Through comparison, it can be concluded that the improved ResNet18 model with transfer learning can minimize the cost function value during the training process. A smaller cost function value indicates better accuracy and robustness of the model.

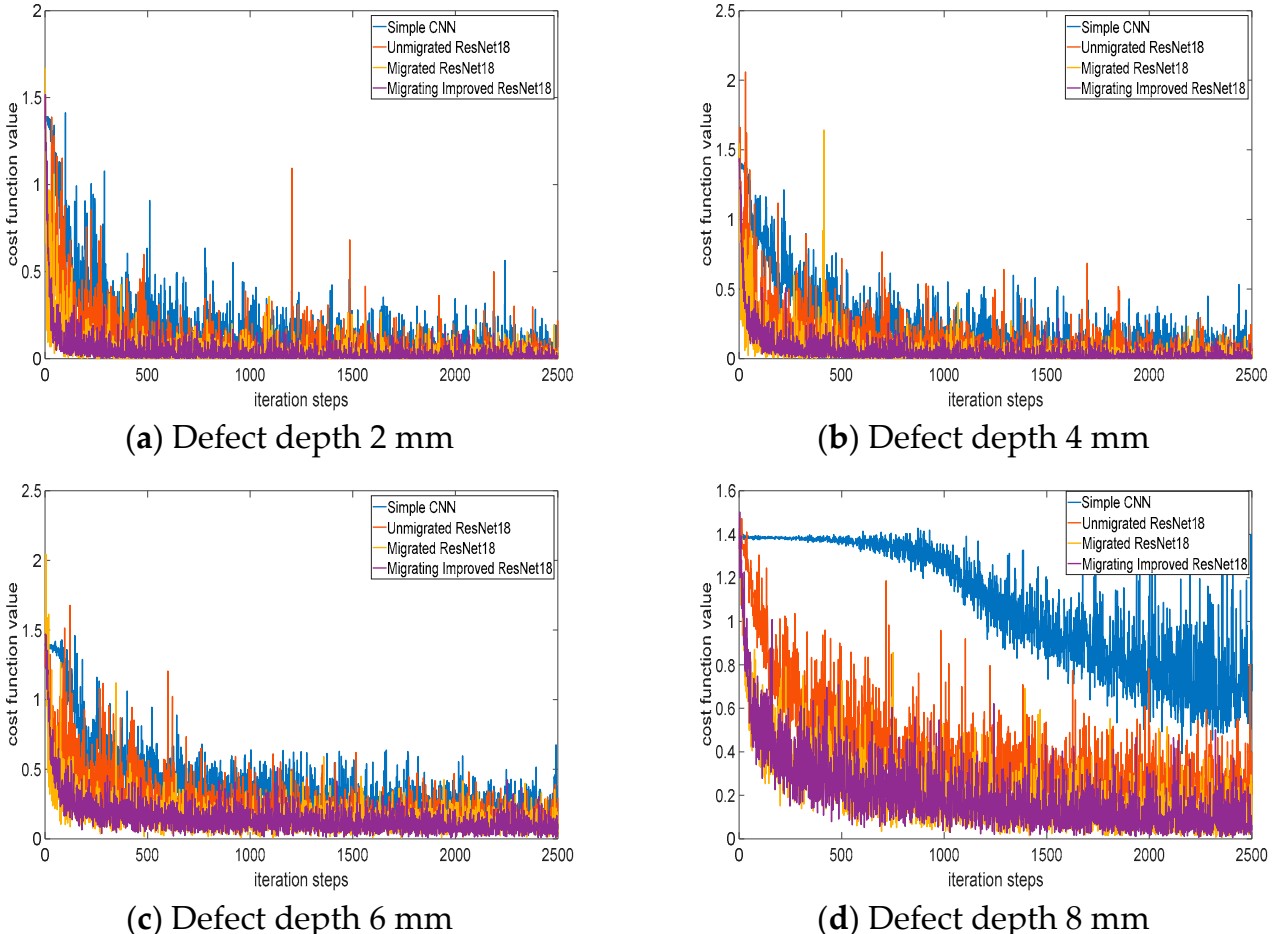

**(a)** Defect depth 2 mm

**(b)** Defect depth 4 mm

**(c)** Defect depth 6 mm

**(d)** Defect depth 8 mm

**Figure 15.** Cost function comparison diagram of 4 scanning detection speeds vs. defects of the same depth.

**Table 3.** Cost function of four scanning detection speeds on defects with different depths.

| Model | 2 mm | 4 mm | 6 mm | 8 mm |
|---|---|---|---|---|
| Improved migration ResNet18 | 0.200 | 0.037 | 0.124 | 0.155 |
| Migrated ResNet18 | 0.200 | 0.040 | 0.125 | 0.159 |
| Unmigrated ResNet18 | 0.290 | 0.044 | 0.14 | 0.190 |
| Simple CNN | 0.520 | 0.076 | 0.20 | 0.399 |

According to Figure 16 and Table 4, it can be seen that the transfer-improved ResNet18 model has the highest accuracy in detecting 2 mm, 4 mm, 6 mm, and 8 mm defects compared with the other three models when the image format is $64 \times 64$. The recognition rate reached the highest at 98.80% when detecting defects with a depth of 2 mm. The recognition rate is the lowest, reaching 93.10%, in the detection of 8 mm depth defects.

From Tables 5–8, it can be seen that the transfer-improved ResNet18 model has the highest overall accuracy, precision, F1, and $R^2$ for the detection of the same type of defects under 4 scanning detection speeds. These results indicate that the improved ResNet18 model through transfer learning has high performance and effectiveness in defect detection tasks, which is very valuable for detecting defects in ferromagnetic materials. This means that the model is able to identify and predict defects more accurately, helping to improve product quality and production efficiency.

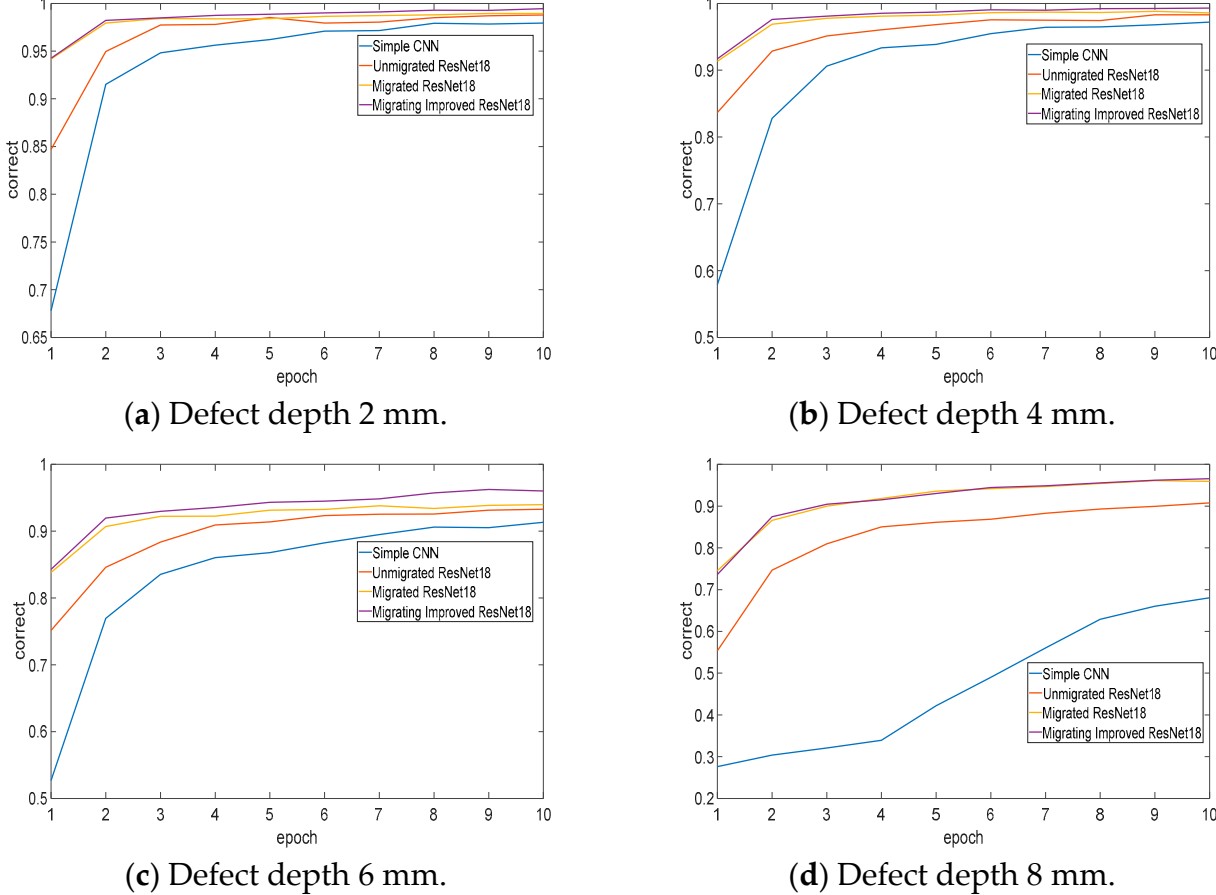

(**a**) Defect depth 2 mm.　　　　(**b**) Defect depth 4 mm.

(**c**) Defect depth 6 mm.　　　　(**d**) Defect depth 8 mm.

**Figure 16.** Comparison of recognition accuracy of defects with different depths by four scanning detection speeds.

**Table 4.** Accuracy (%) of 4 scanning detection speeds for defects with different depths.

| Model | 2 mm | 4 mm | 6 mm | 8 mm |
|---|---|---|---|---|
| Improved migration ResNet18 | 98.8 | 98.7 | 94.3 | 93.1 |
| Migrated ResNet18 | 98.6 | 98.4 | 93.2 | 92.8 |
| Unmigrated ResNet18 | 98.6 | 97.2 | 92.8 | 91.4 |
| Simple CNN | 97.0 | 95.8 | 90.5 | 80.1 |

**Table 5.** Comparison of recognition effect of network model on 2 mm at different scanning detection speeds.

| Model | Accuracy | Precision | F1 | $R^2$ |
|---|---|---|---|---|
| Improved migration ResNet18 | 99.20% | 99.20% | 99.20% | 99.36% |
| Migrated ResNet18 | 98.60% | 98.59% | 98.59% | 98.91% |
| Unmigrated ResNet18 | 98.60% | 98.59% | 98.58% | 98.88% |
| Simple CNN | 97.00% | 97.07% | 96.96% | 97.65% |

**Table 6.** Comparison of recognition effect of network model on 4 mm at different scanning detection speeds.

| Model | Accuracy | Precision | F1 | $R^2$ |
|---|---|---|---|---|
| Improved migration ResNet18 | 98.50% | 98.41% | 98.50% | 98.80% |
| Migrated ResNet18 | 98.40% | 98.38% | 98.38% | 98.72% |
| Unmigrated ResNet18 | 97.20% | 97.22% | 97.19% | 97.47% |
| Simple CNN | 95.80% | 95.89% | 95.78% | 96.72% |

**Table 7.** Comparison of recognition effect of network model on 6 mm at different scanning detection speeds.

| Model | Accuracy | Precision | F1 | $R^2$ |
|---|---|---|---|---|
| Improved migration ResNet18 | 93.30% | 94.30% | 93.24% | 94.65% |
| Migrated ResNet18 | 93.20% | 93.61% | 93.07% | 94.64% |
| Unmigrated ResNet18 | 92.80% | 93.08% | 92.61% | 94.00% |
| Simple CNN | 90.50% | 91.78% | 90.32% | 92.15% |

**Table 8.** Comparison of recognition effect of network model on 8 mm at different scanning detection speeds.

| Model | Accuracy | Precision | F1 | $R^2$ |
|---|---|---|---|---|
| Improved migration ResNet18 | 94.00% | 94.00% | 93.90% | 95.22% |
| Migrated ResNet18 | 92.70% | 92.83% | 92.69% | 94.06% |
| Unmigrated ResNet18 | 89.70% | 90.20% | 89.75% | 89.88% |
| Simple CNN | 69.50% | 69.87% | 67.45% | 69.28% |

## 5. Conclusions

To address the issues of low efficiency in artificial determination of defects in ferromagnetic materials and low accuracy in predicting defect sizes by shallow networks for ferromagnetic materials, this paper proposes the use of an improved convolutional neural network model based on transfer learning. In the improved model, the activation function ReLu is replaced with LeakyReLu to avoid the potential occurrence of neuron "death" phenomenon.

Compared with traditional convolutional neural networks, the improved ResNet18 method has more accurate prediction capability for the size of ferromagnetic material defects and higher training efficiency, with a comprehensive accuracy rate of up to 100%, achieving the best recognition rate for 4 different scanning detection speeds. The experimental results show that the method proposed in this paper provides a comprehensive and better approach for defect prediction in ferromagnetic material leakage detection.

**Author Contributions:** Conceptualization, Y.C., L.L., Z.M. and X.W.; methodology, Y.C. and L.L.; software, L.L.; validation, Y.C., X.W. and L.L.; formal analysis, L.L.; investigation, Y.C., L.L. and J.P.; resources, L.L. and W.Y.; data curation, L.L. and W.Y.; writing—original draft preparation, L.L. and W.Y.; writing—review and editing, Y.C. and Z.M.; visualization, L.L. and J.P.; supervision, Y.C.; project administration, Y.C., X.W. and Z.M.; funding acquisition, Y.C. and X.W. All authors have read and agreed to the published version of the manuscript.

**Funding:** This research was funded by the National Key Research and Development Plan Project, Network Collaborative Manufacturing Integrated Technology and Digital Suite R&D (No. 2020YFB1712400), Henan Province Science and Technology Research Project (No. 232102220025).

**Data Availability Statement:** The data used to support the findings of this study are included within the article.

**Acknowledgments:** This work was supported by the Key Research Project of Higher Education in Henan Province in 2023, "Research on Magnetic Memory Signal Inversion and Interpretation Method for Overhead Pipeline Defects Based on Deep Learning", the Henan Provincial Collaborative Innovation Center of Aerospace Electronic Information Technology, the Key Discipline Group of Aerospace Intelligent Engineering in Henan Province, and the Key Laboratory of General Aviation Technology in Henan Province.

**Conflicts of Interest:** The authors declare no conflict of interest.

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
