# Peer review of "Improved Leakage Detection and Recognition Algorithm for Residual Neural Networks Based on Transfer Learning"

_electronics, doi:10.3390/electronics12204378_

Round 1

Reviewer 1 Report

The paper entitled “Research on Quantitative Identification Algorithm for Defects in Magnetic Flux Leakage Testing Based on Transfer Learning Improved Residual Network” presents the design and training of a convolutional neural network for the detection of defects on metallic pipes.

Although some literature review is presented, some important references are still missing, for example:

Bailey, J., Long, N., & Hunze, A. (2017). Eddy current testing with giant magnetoresistance (GMR) sensors and a pipe-encircling excitation for evaluation of corrosion under insulation. Sensors (Switzerland), 17(10). https://doi.org/https://doi.org/10.3390/s17102229

Chen, J., Hu, W., Cao, D., Zhang, B., Huang, Q., Chen, Z., & Blaabjerg, F. (2019). An imbalance fault detection algorithm for variable-speed wind turbines: A deep learning approach. Energies, 12(14). https://doi.org/10.3390/en12142764

Vankov, Y., Rumyantsev, A., Ziganshin, S., Politova, T., Minyazev, R., & Zagretdinov, A. (2020). Assessment of the condition of pipelines using convolutional neural networks. Energies, 13(3). https://doi.org/10.3390/en13030618

Ma, Q., Tian, G., Zeng, Y., Li, R., Song, H., Wang, Z., Gao, B., & Zeng, K. (2021). Pipeline In-Line Inspection Method, Instrumentation and Data Management. Sensors, 21(11), 3862. https://doi.org/10.3390/s21113862

Zagretdinov, A., Ziganshin, S., Vankov, Y., Izmailova, E., & Kondratiev, A. (2022). Determination of Pipeline Leaks Based on the Analysis the Hurst Exponent of Acoustic Signals. Water, 14(19), 3190. https://doi.org/10.3390/w14193190

Rifai, D., Abdalla, A. N., Ali, K., & Razali, R. (2016). Giant magnetoresistance sensors: A review on structures and non-destructive eddy current testing applications. In Sensors (Switzerland) (Vol. 16, Issue 3). MDPI AG. https://doi.org/10.3390/s16030298

Also, the paper is lacking some explanation on the test object. Only Figure 8 shows some schematics of the pipeline defects. Please add some photographs of the object and the defects used to train the CNN.

Furthermore, no information about the sensor used in the magnetic flux leakage data collection. Please include a photograph of the experimental platform where the sensor and object under test are clearly identified as well as the other instruments.

Author Response

Thanks to the opinions of experts, the relevant references of this paper have been re-searched and sorted out, and the above mentioned papers have been added to the citations of this paper.

For the sensor information used in MFL data collection, This paper adopts Usarek Z,Chmielewski M,Piotrowski L. Reduction of the Velocity Impact on the Magnetic Flux Leakage Signal[J]. The original data in the Journal of Nondestructive Evaluation,2019,38(1). Therefore, no information about the sensors used in MFL data collection is included.

Reviewer 2 Report

I want to commend the authors for the efforts put into the paper. However, I am constrained to recommend Reject for the following reasons: 

1. There is no novelty in the concept and deep learning model

2. There is no system model of the concepts. Thus, it is difficult to situate the major contributions of the authors.

3. The authors highlighted the replacement of RELU with LeakyRelu as a major addition, but that is not new in all CNN or deep learning research.

4. Moreover, there are English errors and template errors with poor referencing style.

5. The title needs adjustment and the entire content of the paper needs to reflect the title "Quantitative" >. This is a strong missing link in the paper. 

6. Find attached the PDF with more corrections. 

Overall, the paper lacks sufficient related works, a convincing introduction, and well-itemized contributions to the body of knowledge. It is of low quality and not recommended to the MDPI electronics journal.  

I have attached the PDF for emphasis. 

1. Too many passive words

2. A native speaker review will help. 

Author Response

Thanks to the opinions of experts.

1.According to the content of the paper, the word quantitative has been deleted, and the title of the paper is changed to "Magnetic leakage detection and Recognition Algorithm based on improved residual neural Network based on transfer learning".
2.This research direction mainly adopts machine learning. In this paper, the magnetic flux leakage signal of ferromagnetic material containing noise is converted into two-dimensional image, and the high-level features of defect signal are obtained with the help of deep learning. Through multi-feature fusion with traditional features, the defect signal is sent into deep learning neural network for recognition and classification. This paper provides a better idea for defect prediction in magnetic flux leakage testing of ferromagnetic materials, and improves the detection performance when defects initially occur.
3.This paper proposes to construct an improved deep learning model with multi-feature fusion. The radial and axial data are fused and sent into the improved model training, which achieves a high recognition rate.
4.The improved model structure differs from the established model structure in that all ReLU activation functions in the ResNet18 network are replaced with LeakyReLU activation functions. When this data is input, it may cause the activation value to be zero when the input value is negative, resulting in the gradient of the neuron becoming zero, and the reverse propagation gradient cannot be transmitted, which may cause some neurons to "die". The LeakyReLU activation function is an improvement on ReLU that introduces a small slope value (usually a small positive number) for a negative input value, so that there is still gradient transfer for a negative value. This avoids the "death" of certain neurons and helps improve the learning ability and performance of the network.
5.The statement and template have been rechanged.

Reviewer 3 Report

The manuscript described a powerful modern approach to deep learning with potentially many different applications and far-reaching implications. The achieved classification accuracies are impressive. Unfortunately, the manuscript is hard to comprehend and evaluate, since the focus and composition of the text and its representation is far from satisfactory. Moreover, the text seems to be hastily written: Many times it includes unfinished thoughts or sentences, comprises long overtreatments of minor details, but lacks important motivations, citations and explanations of core techniques like Transfer Learning / Migration learning (both terminologies are used in parallel) etc.

Major issues:
(1) Title: Omit "Research on". Defects in Magnetic Flux Leakage Testing? Suggestion: "Quantitative Identification Algorithm for Magnetic Flux Leakage Testing Based on Residual Neural Networks Improved by Transfer Learning"  

(2) Please revise and rewrite the Abstract. Eliminate all unfinished sentences. Include numbers of your subject (what defect sizes, scanning speeds) and results (prediction accuracy, detection speed). Clearly state whether simulated, noiseless data or noisy experimental data were used.  

(3) Please be more precise. Examples: (a) Explain, what are radial / axial leakages, show real data of defects and explain, why the curves in Fig. 1 and 2 are (obviously) simulated and on what basis they were simulated (absolute flux numbers are shown). (b) Motivate the parameter settings of your network, Lines 80 to 100.  (c) Is it true, that the improved model structure differs from the established one only in including LeakyReLU activation functions like suggested in Fig. 4? (d) Explain, why the cost function is so important here and what the comparison of the numbers given in Table 3 imply. (e) What are False Negatives in your application? (f) What "defect magnetic flux leakage detection experimental platform" is addressed? Has it been published? What sensor is used to detect magnetic flux leakage data (line 259)? (g) What about the processing times? On what hardware? What detection speeds are achieved (in seconds).

(4) Please make all figure captions as self-explanatory as possible. Examples: (a) Figure 1/2: What is h, why it is representative for the defect size (area, volume?); what distance? (b) Figure 3: Much of the Features extraction (Chapter 2.2.2) would be needed here to better understand the details. "Simple CNN structure" (in general, your application?) is definitely not sufficient. (c) Figure 15 is unclear: What is shown here? Detection speeds?

(4) Please, be more concise. Reconsider the proportions of the text body. Add references wherever new terminology is used (e.g. Transfer learning. ResNet18)

(5) Please, improve the graphical representation. Examples: (a) The numbers in the confusion matrices (Figs 12, 14) are far too small. (b) Why 4 Tables (Table 5-8) are needed to show numbers, which are better highlighted by curves?

To summarize: In a recommended, revised version of the manuscript, after having streamlined the text to improve readability and comprehension, the generalization capacity of the novel approach (applicable not only for oil and gas pipelines) should be highlighted. 

Please make sure that you consult the different stages of textwriting with all the co-authors and be advised to circulate the manuscript among colleagues to seek their advice before submitting it to a Journal.

(1) Avoid repetitions, e.g. line 45.

(2) Avoid unfinished sentences (starting from the Abstract)

(3) Please check the following (and other) phrases for sense and meaning:
"signal to grayscale image", "neuron death", "Under four different inspection speeds, the recognition rate for the same defect is the highest", "better comprehensive performance method"

Author Response

Thanks to the opinions of experts.
1.The title of the paper has been revised to "Improved Leakage Detection and Recognition Algorithm for Residual Neural Networks based on Transfer Learning ".
2.The summary has been rewritten to eliminate all unfinished sentences. The type of experimental data is clearly stated.
3.(a) Radial magnetic leakage refers to the horizontal component of MFL, and axial magnetic leakage refers to the vertical component of MFL. The curves in Figures 1 and 2 were simulated to study the effect of defect depth on the MFL signal, based on the rectangular model magnetic dipole model.

(b) The parameter Settings for the incentive network on lines 80 to 100 have been written in detail.

(c) The improved model structure differs from the established model structure by replacing all ReLU activation functions in the ResNet18 network with LeakyReLU activation functions. The input of this data may cause the activation value to be zero when the input value is negative, resulting in the gradient of the neuron becoming zero, and the reverse propagation gradient cannot be transmitted, which may cause some neurons to "die". The LeakyReLU activation function is an improvement on ReLU that introduces a small slope value (usually a small positive number) for a negative input value, so that there is still gradient transfer for a negative value. This avoids the "death" of certain neurons and helps improve the learning ability and performance of the network.

(d) The cost function is the average of all sample errors, which is used to measure the difference between the predicted results of the model and the actual label. It can also be called a loss function or an objective function. In this article, cross entropy is used as a cost function. Cross-entropy can measure the difference between the model output and the actual label, and can provide effective gradient information in the process of model optimization, so that the model can be updated quickly and effectively. If the classification result of the model is very different from the actual label, the value of cross entropy will be large, and the smaller the difference between the prediction result and the label, the smaller the value of cross entropy will be, that is, the smaller the cost function value, the better the fit of the trained model.

(e) FN (false negative) : The number of negative samples wrongly classified by intrusion detection model in abnormal behavior. In this paper, FN is mainly caused by the addition of Gaussian noise to the data and the rapid detection rate that causes data variation and increases the outliers in the data, resulting in the model incorrectly judging positive samples as negative samples for inaccurate samples, thus generating false negatives.

(f) Usarek Z,Chmielewski M,Piotrowski L. Reduction of the Velocity Impact on the Magnetic Flux Leakage Signal is adopted in this paper. [J]. The original data in the Journal of Nondestructive Evaluation,2019,38(1). The MFL tool is used to scan the detection data of the test board made of S355 steel grade. The detection tool is a measurement platform composed of three magnetizers, a digital encoder and a measurement module. It uses a linear Hall effect sensor A1324 to measure flux leakage.
4.(a)h stands for defect depth, whose model and formula have been marked in the paper.
(b) A feature extraction detail section has been added.
(c) The classification of four different detection speeds for the same defect is shown, and the distribution of the relationship between the number of iteration steps and the cost function of the four models has been updated.
5. Relevant literature on transfer learning and ResNet18 has been cited, and some basic introductions to transfer learning and RseNet18 have been deleted, which have been marked in the paper.

6.(a) The numbers in Figure 12 and 14 of the confusion matrix have been increased.

(b) The four tables can be compared more vividly to show that the four parameters of the improved ResNet18 are higher than those of the other three, indicating that the model has been successfully improved. As can be seen from Table 5-8, the migrated and improved ResNet18 model has the highest accuracy, precision, F1 and R2 overall for the same type of defect detection at 4 detection speeds. These results show that the improved ResNet18 model has high performance and effect in the task of defect detection, which is very valuable for the detection of ferromagnetic material defects. This means models can more accurately identify and classify defects, helping to improve product quality and production efficiency.

Round 2

Reviewer 2 Report

 want to thank the authors for ensuring that all previous issues were addressed. However, the following need to be corrected: 

1. Lines 338, 341, 413, 426, 427, 428, and 429 should be R2, not R2

2. Wrong referencing still persists. 

[6] , [7], and [8] are wrong and the authors are confusing. 

3. Kinldy delete [j], [C], and [D] from the referencing in [10], [13], [15], [16], [17], and [18].

4. In reference [14], 'survey' can not be the name of an author. kindly use the correct MDPI template for all referencing

5. In [19], provide spacing between author names.

In summary, kindly edit properly and properly a CLEAN copy of the paper for clarity. 

Author Response

Thanks for the valuable advice from the experts, I have revised the opinions given by the experts one by one. To make it easier for experts to read, I have highlighted the changes in red.

     1.Lines 338, 341, 413, 426, 427, 428, and 429 should be R2, not R2

Answer: R2 in the article has been modified to R², MT to MT, and LR to LR.

  1. Wrong referencing still persists. 

[6] , [7], and [8] are wrong and the authors are confusing. 

Answer: The citations [6], [7] and [8] have been revised.

  1. Kinldy delete [j], [C], and [D] from the referencing in [10], [13], [15], [16], [17], and [18].

Answer: All references have been reviewed and revised.

  1. In reference [14], 'survey' can not be the name of an author. kindly use the correct MDPI template for all referencing

Answer: I have modified it using the correct MDPI template.

  1. In [19], provide spacing between author names.

Answer: Space has been provided between author names in [19].

Reviewer 3 Report

Dear Authors,

thank you for undertaking the effort of significantly improving the manuscript and re-writing parts of it. Some remarks remain:

1) Abstract: The Abstract is still very general. Please provide numbers (size, depth) of the defect to be classified and the classification rates, at least for the best method.

2) Introduction: Lines 56ff are unclear and must be improved: "It (CNNs? plural) uses the (?) model to extract features, allowing it (?) to extract high-dimensional features (?) and overcome the limitations of manual features (?)." What are "manual features"? Please describe and show the "leakage defect data features" (line 66, not a good expression) in order to give the readers an impression, what defects are to be detected and classified.

3) Basic principles

Line 70-72 and lines 80-89 are unclear and not well written. Maybe a sketch would help to imagine the situation (depth, width, distance, axial/radial data?) Are the described distances between the valleys the features? Please indicate clearer in the figure captions of Figs. 1/2 what parameter values were used to calculate these simulated curves and what features are drawn from it. (Btw: It should be made clear what h is, without the need to read the text.)

Figure 3: Why 64 x 64 images as inputs? Was this motivated before? It is suggested to refer to Figure 6 or to present it and the description for pre-processing the data earlier in the text.

Figure 4: Its unclear from the figure, what has been improved. A said before, it looks, as if the LeakyReLU insertions were the only improvements, but this is not transfer learning. Please re-consider the figure presentation. 

Figure 8: Please enlarge the graphs and use less circles, to make the features  more visible. (Caption: "feature maps"!)

Figures 9/10: Please motivate the improvements. What exactly is improved and why?

Figure 11: What is the statement/message? What is Figure 11 used for? 

Figure 12 and Figure 14 d) are identical. It would be sufficient to show 14 d) and cancel Fig. 12.

Caption 4.4. It is unclear why adding noise should have something to do with "detection speed". What is the "speed" in Figure 15 and 16? If you mean different algorithms, then "speed" is definitely not the right term.  Please give numbers for detection speeds (or times to learn to features) in seconds, for the computer configuration you used.

Table 2 and Table 4: One digit after the comma is enough.

Summary: Please reconsider parts of the manuscript in terms of a structured comprehensive presentation of a) the subject, b) motivations behind core decision regarding e.g. the architecture and feature selection and c) the graphical presentation.

Do not say "speeds" when you mean "algorithms". Would your proposed algorithms be the fastest?

Author Response

Thanks for the valuable advice from the experts, I have revised the opinions given by the experts one by one. To make it easier for experts to read, I have highlighted the changes in red.

1) Abstract: The Abstract is still very general. Please provide numbers (size, depth) of the defect to be classified and the classification rates, at least for the best method.

Answer: What has been added: " The results demonstrated that the improved ResNet18 network model, after transfer learning, achieved 100% prediction accuracy for all 10,000 grayscale images generated with defect lengths of 50mm, width of 2mm, and depths of 2mm, 4mm, 6mm, and 8mm. Moreover, the prediction accuracies for the quasi-static, slow, compensated fast, and fast scanning speeds were respectively 99.20%, 98.50%, 93.30%, and 94.00% for defect depths of 2mm, 4mm, 6mm, and 8mm. These accuracies surpass those of other models, demonstrating the significant improvement in prediction accuracy achieved by this method ".

2) Introduction: Lines 56ff are unclear and must be improved: "It (CNNs? plural) uses the (?) model to extract features, allowing it (?) to extract high-dimensional features (?) and overcome the limitations of manual features (?)." What are "manual features"? Please describe and show the "leakage defect data features" (line 66, not a good expression) in order to give the readers an impression, what defects are to be detected and classified.

Answer: It in the article refers to CNN's use of convolutional neural network models to extract features, and artificial features refer to features designed and selected by humans for identifying and classifying objects. However, there are some limitations to artificial feature extraction, including the fact that it requires a lot of time and expertise, and the expressive power is limited. CNN uses convolutional neural network model for feature extraction to extract features of higher dimensions. Here, high-dimensional features are represented as feature vectors with a large number of elements or indicators. Compared with low-dimensional features, high-dimensional features can provide more information and richer representation capabilities. In the magnetic leakage detection of ferromagnetic material defects, permanent magnets are used to excite the ferromagnetic material, and a closed magnetic circuit is formed between the armature, air gap and ferromagnetic material. After the saturation excitation of the ferromagnetic material, the magnetic induction intensity inside the ferromagnetic material without defects remains stable, that is, the magnetic field line is evenly distributed in the ferromagnetic material, while the magnetic permeability decreases sharply at the defect site. At this time, the magnetic field line will overflow from the defect surface of the ferromagnetic material, thus forming a leakage magnetic field in the air on the surface of the ferromagnetic material. The strength of the leakage magnetic field is directly related to the distribution and excitation intensity, defect type and defect size. When the excitation intensity is determined, the damage can be quantitatively identified according to the distribution of the leakage magnetic field. The magnetic sensor is used to obtain the above leakage magnetic field signal, and the corresponding defect width, depth and other characteristics can be extracted. A diagram has been drawn to represent the leakage defect data characteristics, as shown in Figure 1.

3) Basic principles

Line 70-72 and lines 80-89 are unclear and not well written. Maybe a sketch would help to imagine the situation (depth, width, distance, axial/radial data?) Are the described distances between the valleys the features? Please indicate clearer in the figure captions of Figs. 1/2 what parameter values were used to calculate these simulated curves and what features are drawn from it. (Btw: It should be made clear what h is, without the need to read the text.)

Answer: A sketch has been drawn to express the magnetic flux leakage signals of the same length, width and depth and the axial and radial data features. Distance refers to the coordinates in the X-axis direction with the rectangular defect center as the coordinate origin. The distance between the valleys is characteristic. It has been more clearly indicated in the figure title of Figure 1/2 (now Figure 3/4) that these simulation curves are made with defect width n=5mm, lift distance y=5mm, defect depth h of 2mm, 4mm, 6mm, 8mm in turn. Under the same conditions, the larger the defect depth is, the larger the difference between the peak value and the valley value of the radial MFL signal data will be, and the peak value of the axial MFL signal data will be. (The h in the title of Figure 1/2 (now Figure 3/4) has been replaced by Depth to make it easier to read and understand, as shown in Figure 3/4.)

Figure 3: Why 64 x 64 images as inputs? Was this motivated before? It is suggested to refer to Figure 6 or to present it and the description for pre-processing the data earlier in the text.

Answer: In order to improve the classification effect, the size of the image in this study varies according to the amount of signal data, and a larger image size can improve the classification result. In this paper, the amount of signal data is large, and when the size of the generated gray image is set to 64×64, a total of 10,000 gray images can be generated, which can meet the training requirements of convolutional neural networks. Therefore, it is most appropriate to set the input image size to 64×64.

Figure 4: Its unclear from the figure, what has been improved. A said before, it looks, as if the LeakyReLU insertions were the only improvements, but this is not transfer learning. Please re-consider the figure presentation. 

Answer: Figure 4 (now Figure 6) has been redrawn. The training parameters of the ResNet18 network in the CIFAR10 dataset are migrated and the ReLU activation function in the ResNet18 model is replaced with the Leaky ReLU function. Then the gray-scale image data set generated in this paper is fed into the migrated and improved ResNet18 network model for training classification.

Figure 8: Please enlarge the graphs and use less circles, to make the features  more visible. (Caption: "feature maps"!)

Answer: Figure 8 (now Figure 10) has been enlarged to make the features more visible. I have added content to the article stating that under the same conditions, the deeper the defect depth, the darker the generated grayscale image.

Figures 9/10: Please motivate the improvements. What exactly is improved and why?

Answer: Figure 9/10 (now Figure 11/12) shows the details of the changes in the ResNet18 model. It has been marked with different colors. Thanks to the expert's advice, it has been changed.

Figure 11: What is the statement/message? What is Figure 11 used for? 

Answer: Figure 11 was originally intended to show that the loss value of the model is getting smaller and smaller, and the recognition rate of the model is getting better and better. Figure 13 now has the same central meaning as Figure 11, and Figure 11 has been removed.

Figure 12 and Figure 14 d) are identical. It would be sufficient to show 14 d) and cancel Fig. 12.

Answer: Thanks to expert advice, Figure 12 has been removed.

Caption 4.4. It is unclear why adding noise should have something to do with "detection speed". What is the "speed" in Figure 15 and 16? If you mean different algorithms, then "speed" is definitely not the right term.  Please give numbers for detection speeds (or times to learn to features) in seconds, for the computer configuration you used.

Answer: Since convolutional neural network learning requires a large amount of data, adding Gaussian noise to experimental data obtained by scanning detection speed can not only meet the training requirements of deep learning, but also restore the scene of field detection better by adding a certain amount of noise, because experimental data obtained by scanning detection speed are more susceptible to noise during field detection of magnetic flux leakage detection. Therefore, the increase of noise can better meet the needs of field detection data, and highlight the excellent performance of the improved ResNet18 model after migration. The "speed" in Figures 15 and 16 refers to the speed of scanning the ferromagnetic material with the instrument. Since the four detection interval speeds, quasi-static, slow, compensated fast, and fast, were provided in the cited literature, the exact speed could not be determined, so we apologize to the experts here.

Table 2 and Table 4: One digit after the comma is enough.

Answer: Thanks to expert advice, Forms 2 and 4: one decimal place have been retained.